# Multiplicative Weights Updates with Constant Step-Size in Graphical Constant-Sum Games

**Yun Kuen Cheung** *
Singapore University of Technology and Design
Singapore
yunkuen_cheung@sutd.edu.sg

## Abstract

Since Multiplicative Weights (MW) updates are the discrete analogue of the continuous Replicator Dynamics (RD), some researchers had expected their qualitative behaviours would be similar. We show that this is false in the context of graphical constant-sum games, which include two-person zero-sum games as special cases. In such games which have a fully-mixed Nash Equilibrium (NE), it was known that RD satisfy the permanence and Poincaré recurrence properties, but we show that MW updates with any constant step-size $\varepsilon > 0$ converge to the boundary of the state space, and thus do not satisfy the two properties. Using this result, we show that MW updates have a regret lower bound of $\Omega(1/(\varepsilon T))$, while it was known that the regret of RD is upper bounded by $\mathcal{O}(1/T)$.

Interestingly, the regret perspective can be useful for better understanding of the behaviours of MW updates. In a two-person zero-sum game, if it has a unique NE which is fully mixed, then we show, via regret, that for any sufficiently small $\varepsilon$, there exist at least two probability densities and a constant $Z > 0$, such that for any arbitrarily small $z > 0$, each of the two densities fluctuates above $Z$ and below $z$ infinitely often.

## 1 Introduction

The concept of Nash Equilibrium (NE) has been central in game theory. The existential proof of Nash [20] is non-constructive, while the definition of NE itself is also *static*, both of which shed no insight how NE can be computed or reached. In turn, lots of researchers have devoted efforts to *justify* the concept of NE by providing algorithms/dynamics which might compute/reach a NE. Among them, Multiplicative Weights (MW) updates have drawn a lot of attention, due to its simplicity and naturalness, and perhaps more importantly, its distributive implementability[2] which is essential in games we observe in reality, where information communicated between players is often very limited. MW updates have also made profound impacts in algorithm design; see [1] for details. However, various PPAD-hardness and communication complexity results [8, 6, 13] serve as strong indicators that no efficient algorithm/dynamic, MW updates included, can efficiently compute/reach NE for general games. But can MW updates do so for interesting sub-families of games?

The best we could hope for is pointwise convergence toward a NE, but it is known not to hold even in the simplest scenario of two-person zero-sum games. A weaker notion of convergence, called *empirical convergence* (i.e., the average of the time series history converges), has been sought. While this notion might seem less natural, it is interesting from the perspective of statistics; this

notion, coined as "ergodic convergence" in the study of dynamical systems, is central in a branch of mathematics called *ergodic theory*, where its initial development was motivated by the study of *time-average* behaviours of systems of interest in statistical physics. Freund and Schapire [10] showed that MW updates converge to NE empirically in any two-person zero-sum game using the notion of regret; their analysis also yields a simple and beautiful proof of John von Neumann's Minimax Theorem. Daskalakis and Papadimitriou [9] and Cai and Daskalakis [4] generalized to graphical constant-sum games and separable zero-sum games.

It should be noted that, however, their precise result is actually the following: if the finishing time $T$ is known a priori[3], then there is a step-size $\varepsilon$, which depends on $T$, such that the regret at time $T$ is $\mathcal{O}(1/\sqrt{T})$, hence the empirical average at time $T$ forms an $\mathcal{O}(1/\sqrt{T})$-approximate NE. When $\varepsilon$ is a fixed constant and $T \nearrow \infty$, their analyses can only show a regret bound of $\mathcal{O}(\varepsilon)$, and it is not clear whether empirical convergence toward NE occurs. Recently there are several work showing distributed dynamics achieve $o_T(1)$ regret in two-person zero-sum games [7, 24] and general games [28], but all of them require the step-size to be diminishing or adaptive. However, in some applications of biological system models, e.g., population dynamics (see [26, 15]) and evolution [5], diminishing or adaptive step-size is unnatural. Also, from an algorithmic perspective, it is natural to ask what happens if the step-size is constant, and if the use of diminishing or adaptive step-size is unavoidable. Indeed, in last year, Palaiopanos et al. [21] have addressed the same question in the context of congestion games.

In this paper, we study MW updates with constant step-size in graphical constant-sum games. It was known that Replicator Dynamics (RD), the continuous analogue of MW updates, in such games are permanent [14], i.e., all probability densities are bounded away from zero throughout. Also, RD in such games satisfy the Poincaré recurrence property [23, 18]. In algorithm design and modelling of biological systems, continuous-time dynamics like RD are hardly feasible, so we are interested in the behaviours of MW updates. Due to the analog between MW updates and RD, it seems natural to expect MW updates could retain the above good properties of RD. Unfortunately, this is not true. We show that MW updates in such games do not satisfy permanence and Poincaré recurrence properties; indeed, we show that MW updates converge toward the boundary of the state space (unless the starting point is NE), which is stronger than non-satisfaction of the above two properties.

We note a recent independent work of Bailey and Piliouras [2], who show the same result in a more general setting: first, they generalize to discrete Follow-The-Regularized-Leader (FTRL) dynamics, and second, they allow the step-sizes to be mildly decreasing. Precisely, they show that if a FTRL dynamic guarantees every update must stay in the interior of the state space (MW is an example of such FTRL dynamic), then the dynamic converges toward the boundary; otherwise, they show that the FTRL dynamic gets arbitrarily close to the boundary infinitely often (this is weaker than "convergence toward the boundary"). In our paper, we proceed further by using the result to get a better understanding of the behaviours of MW updates and the regret, as we will discuss next.

We show a regret lower bound of $\Omega(1/(\varepsilon T))$ plus a positive term, which is proportional to the average variance of payoffs among strategies over time. This lower bound should be compared to RD's regret upper bound $\mathcal{O}(1/T)$ by Sorin [27, Theorem 3.1].[4] Mertikopoulos et al. [18] generalized Sorin's upper bound to continuous FTRL dynamics.

Interestingly, the regret perspective can be useful for better understanding of the behaviours of MW updates. In a two-person zero-sum game, if it has a unique NE which is fully mixed, then we show, via regret, that for any sufficiently small $\varepsilon$, there exist at least two probability densities and a constant $Z > 0$, such that for any arbitrarily small $z > 0$, each of the two densities fluctuates above $Z$ and below $z$ infinitely often.

**Continuous vs. Discrete Dynamics.**    As we will see, from a high-level perspective, we are exploiting the interplay between continuous dynamics and their analogous discrete dynamics. Such interplay has been exploited before in other contexts; see, for instance, Sorin [27], Kwon and Mertikopoulos [17] and Benaïm [3]. Briefly speaking, in [27, 17], they showed that the disparity between the two dynamics is under control up to a certain time by choosing a suitable time-dependent step-size. Thus,

if a good property holds for the continuous dynamic (which is often easier to show in the continuous time setting), it might carry over to the discrete analogue, sometimes with a depreciation due to the disparity. In the contexts we study, we show that the disparity *must* accumulate indefinitely, and eventually lead to very different qualitative behaviours between RD and MW updates.

**Other Related Work.**   The long term (asymptotic) behaviours of learning/evolution dynamics in games, biological and other systems have attracted attention from researchers across multiple disciplines for decades; see the text of Hofbauer and Sigmund [15] for an extensive summary. A large number of work have focused on continuous dynamics. Even in simple games like Paper-Rock-Scissors or small three-player graphical games, learning dynamics exhibit rich and sometimes surprising long term behaviours; see, for instance, [29, 11, 12, 25, 16, 22, 19]. Discrete dynamics are more relevant from an algorithmic perspective, and certainly deserve more attention in this algorithmic era. Our work suggests that the long term behaviours of continuous dynamics and their discrete counterparts can be very different, and a better understanding of such behaviours might shed insights on game-theoretic benchmarks such as regret.

## 2   Preliminary

**Types of Games, and Nash Equilibrium.**   In a general bimatrix game with two players, suppose Players A and B have strategy sets $S_A$ and $S_B$ respectively. The game is depicted by a bimatrix $\mathbf{M} = [(a_{ij}, b_{ij})]_{i \in S_A, j \in S_B}$: when Player A chooses strategy $i$ and Player B chooses strategy $j$, $a_{ij}, b_{ij}$ are the payoffs to Players A and B respectively. Such a game is called **two-person constant-sum game** if for all $i \in S_A, j \in S_B$, $a_{ij} + b_{ij} = C$ for some real number $C$; such a game is called **two-person zero-sum game** if it is a two-person constant-sum game with $C = 0$.

In a game with $m$ players, we number the players by $1, 2, \cdots, m$, and let $S_i$ denote the strategy set of Player $i$, and $n_i := |S_i|$. For any $\mathbf{s} = (s_1, s_2, \cdots, s_m) \in \times_{i=1}^m S_i$, where $s_i$ is the choice of strategy of Player $i$, let $u_i(\mathbf{s})$ denote the payoff to Player $i$. Such a game is called **separable zero-sum multiplayer game** if for any $\mathbf{s} = \in \times_{i=1}^m S_i$, $\sum_{i=1}^m u_i(\mathbf{s}) = 0$.

A game with $m$ players is a **graphical polymatrix game** if the game is defined as follows: on an undirected graph $G = ([m], E)$, each edge $(i, j) \in E$ corresponds to a bimatrix game between Players $i$ and $j$ with strategy sets $S_i$ and $S_j$ respectively. It is worth noting that the strategy set of a Player $i$ in different bimatrix games is the same, and every time she plays the game, she must choose the same strategy for all these bimatrix games. Such a game is a **graphical constant-sum game** if the bimatrix game corresponded by every edge is a two-person constant-sum game (different bimatrix games may have different constants $C$).

**Theorem 1** ([4]). *Every separable zero-sum multiplayer game can be transformed into a graphical constant-sum game, while preserving all the payoffs.*

Due to the above theorem, we focus on developing our results on graphical constant-sum games; all these results automatically carry over to separable zero-sum multiplayer games. Also, by suitable scaling, we can assume that the payoff to every player always lie within the interval $[-1, +1]$.

A mixed strategy of a Player $i$ is a probability distribution over her strategy set, represented by a vector $\mathbf{x}_i \in \mathbb{R}^{n_i}$ with $\sum_{j \in S_i} x_{ij} = 1$, where $x_{ij}$ is the probability density that strategy $j$ is chosen. When each Player $i$ chooses a mixed strategy $\mathbf{x}_i$ independently, the payoff function extends naturally by $u_i(\mathbf{x}_1, \mathbf{x}_2, \cdots, \mathbf{x}_m) = \mathbb{E}_{\mathbf{s} \sim \times_{j=1}^m \mathbf{x}_j}[u_i(\mathbf{s})]$. The **boundary** of the mixed strategy space is $\cup_{i, j \in S_i}\{\mathbf{x} \mid x_{ij} = 0\}$.

We say $(\mathbf{x}_1, \mathbf{x}_2, \cdots, \mathbf{x}_m)$ is a **Nash equilibrium** (NE) if no player can change her mixed strategy for achieving a higher payoff. Precisely, for any Player $i$, let $\mathbf{x}_{-i}$ denote the mixed strategies chosen by all players other than Player $i$, then for any mixed strategy $\mathbf{x}_i'$ of Player $i$, $u_i(\mathbf{x}_i', \mathbf{x}_{-i}) \leq u_i(\mathbf{x}_i, \mathbf{x}_{-i})$. A NE is said to be **fully mixed** if no probability density in any of the $\mathbf{x}_i$ is zero. For any Player $i$, let $\mathbf{e}_j$ denote a pure strategy $j$ of her (i.e., a mixed strategy with probability one on strategy $j$).

**Replicator Dynamic and Multiplicative Weights Updates.**   In a game with $m$ players where each player employs a Replicator Dynamic (RD), each Player $i$ maintains a mixed strategy $\mathbf{x}_i(t)$ which is updated continuously with time $t$. Let $\mathbf{x}(t) = (\mathbf{x}_1(t), \mathbf{x}_2(t), \cdots, \mathbf{x}_m(t))$. The update rule is given

by a differential equation system, for each strategy $j \in S_i$,

$$\frac{\mathrm{d}}{\mathrm{d}t} x_{ij}(t) \ = \ x_{ij}(t) \cdot [u_i(\mathbf{e}_j, \mathbf{x}_{-i}(t)) \ - \ u_i(\mathbf{x}(t))].$$

If times are discrete at non-negative integers, and if each Player $i$ employs a Multiplicative Weights (MW) updates with step-size $\varepsilon_i > 0$, the update rule is

$$x_{ij}(t+1) \ = \ \frac{x_{ij}(t) \cdot \exp\left(\varepsilon_i \cdot u_i(\mathbf{e}_j, \mathbf{x}_{-i}(t))\right)}{\sum_{k \in S_i} x_{ik}(t) \cdot \exp\left(\varepsilon_i \cdot u_i(\mathbf{e}_k, \mathbf{x}_{-i}(t))\right)}.$$

It is well-known that MW updates are discrete analogue of RD.

Throughout this paper, we always assume that for all $i \in [m]$ and $j \in S_i$, every starting density $x_{ij}(0)$ is strictly positive, i.e., $\mathbf{x}(0)$ is fully mixed. Also, in all our results, we always assume $\varepsilon_i \leq 1/4$.

The **regret** of Player $i$ is

$$\frac{1}{T}\left[\left(\max_{j \in S_i} \int_0^T u_i(\mathbf{e}_j, \mathbf{x}_{-i}(t))\,\mathrm{d}t\right) - \int_0^T u_i(\mathbf{x}(t))\,\mathrm{d}t\right] \quad \text{for continuous-time model, } T > 0;$$

$$\frac{1}{T}\left[\left(\max_{j \in S_i} \sum_{t=0}^{T-1} u_i(\mathbf{e}_j, \mathbf{x}_{-i}(t))\right) \ - \ \sum_{t=0}^{T-1} u_i(\mathbf{x}(t))\right] \quad \text{for discrete-time model, } T \in \mathbb{N}.$$

## 3 Permanence and Poincaré Recurrence

We first present two prior results concerning RD in general games and graphical constant-sum games.

**Theorem 2** ([27]). *In any multiplayer game where the payoff function to Player $i$ is Lebesgue integrable in the mixed strategies of all players and the game parameters, if Player $i$ employs RD, while the game parameters and mixed strategies of all other players are measurable functions of time, then for any $T > 0$, the regret of Player $i$ is at most $\frac{1}{T} \cdot \max_{j \in S_i} \ln \frac{1}{x_{ij}(0)}$.*

**Theorem 3** ([23]; see also [15, 18]). *If a graphical constant-sum game admits a fully-mixed NE $\mathbf{x}^*$ and all players employ RD, then for any fully mixed starting point $\mathbf{x}(0)$, $H(t) := -\sum_{i=1}^m \sum_{j \in S_i} x_{ij}^* \cdot \ln(x_{ij}(t))$ is a constant for all $t \geq 0$. Consequently, for all $t \geq 0$, $x_{ij}(t) \geq \exp\left(-H(0)/x_{ij}^*\right) > 0$, so the system is permanent (i.e., all $x_{ij}$'s are bounded away from zero throughout), and the $\omega$-set of the dynamic is bounded away from the boundary. Also, the dynamic satisfies the Poincaré recurrence property.*

For MW updates, to cope with the scenarios where different players use different step-sizes, we make a slight modification of the Hamiltonian function $H$ in [23]:

$$H(t) \ := \ -\sum_{i=1}^m \frac{1}{\varepsilon_i} \cdot \sum_{j \in S_i} x_{ij}^* \cdot \ln(x_{ij}(t)).$$

**Lemma 4.** *If a graphical constant-sum game admits a fully-mixed NE $\mathbf{x}^*$ and all players employ MW updates, then $H(t+1) \geq H(t)$. More specifically, let $V_i(t)$ denote*

$$\sum_{k \in S_i} x_{ik}(t) \cdot \left(\exp\left(\varepsilon_i \cdot u_i(\mathbf{e}_k, \mathbf{x}_{-i}(t))\right) - 1\right)^2 \ - \ \left(\sum_{k \in S_i} x_{ik}(t) \cdot \left(\exp\left(\varepsilon_i \cdot u_i(\mathbf{e}_k, \mathbf{x}_{-i}(t))\right) - 1\right)\right)^2,$$

*then $\sum_{i=1}^m \frac{V_i(t)}{\varepsilon_i} \ \geq \ H(t+1) - H(t) \ \geq \ \frac{1}{4}\sum_{i=1}^m \frac{V_i(t)}{\varepsilon_i}$.*

Before proving the lemma, we note that $V_i(t)$ is indeed the variance of the following random variable, and thus is always non-negative: the random variable realizes the value $\left(\exp\left(\varepsilon_i \cdot u_i(\mathbf{e}_k, \mathbf{x}_{-i}(t))\right) - 1\right)$ with probability $x_{ik}(t)$, for all $k \in S_i$. Moreover, if $\mathbf{x}_i(t)$ is fully mixed, then $V_i(t)$ is zero if and only if $u_i(\mathbf{e}_k, \mathbf{x}_{-i}(t))$ is identical for all $k \in S_i$.

*Proof.* We first expand $H(t+1) - H(t) = -\sum_{i=1}^{m} \frac{1}{\varepsilon_i} \cdot \sum_{j \in S_i} x_{ij}^* \cdot \ln \frac{x_{ij}(t+1)}{x_{ij}(t)}$ as follows:

$$= -\sum_{i=1}^{m} \frac{1}{\varepsilon_i} \cdot \sum_{j \in S_i} x_{ij}^* \cdot \left[ \varepsilon_i \cdot u_i(\mathbf{e}_j, \mathbf{x}_{-i}(t)) - \ln \left( \sum_{k \in S_i} x_{ik}(t) \cdot \exp \left( \varepsilon_i \cdot u_i(\mathbf{e}_k, \mathbf{x}_{-i}(t)) \right) \right) \right]$$

$$= -\sum_{i=1}^{m} \left[ \left( \sum_{j \in S_i} x_{ij}^* \cdot u_i(\mathbf{e}_j, \mathbf{x}_{-i}(t)) \right) - \frac{1}{\varepsilon_i} \cdot \underbrace{\ln \left( \sum_{k \in S_i} x_{ik}(t) \cdot \exp \left( \varepsilon_i \cdot u_i(\mathbf{e}_k, \mathbf{x}_{-i}(t)) \right) \right)}_{\mathcal{L}} \right];$$

the final equality holds since for each $i \in [m]$, $\sum_{j \in S_i} x_{ij}^* = 1$.

Noting that each $(\exp (\varepsilon_i \cdot u_i(\mathbf{e}_k, \mathbf{x}_{-i}(t))) - 1)$ is within the interval $[e^{-\varepsilon_i} - 1, e^{\varepsilon_i} - 1]$, and noting that in this interval the function $\ln(1+y) + y^2/4$ is concave but the function $\ln(1+y) + y^2$ is convex, by the Jensen's inequality, we have

$$\sum_{i=1}^{m} V_i(t) \geq \mathcal{L} - \sum_{k \in S_i} x_{ik}(t) \cdot \varepsilon_i \cdot u_i(\mathbf{e}_k, \mathbf{x}_{-i}(t)) \geq \sum_{i=1}^{m} \frac{V_i(t)}{4}. \tag{1}$$

Thus, $\displaystyle\sum_{i=1}^{m} \frac{V_i(t)}{\varepsilon_i} \geq H(t+1) - H(t) - \sum_{i=1}^{m} \sum_{k \in S_i} (x_{ik}(t) - x_{ik}^*) \cdot u_i(\mathbf{e}_k, \mathbf{x}_{-i}(t)) \geq \frac{1}{4} \sum_{i=1}^{m} \frac{V_i(t)}{\varepsilon_i}$.

By following the proof of Theorem 3 in [23], one can show that the above double summation is zero. We defer this part of the proof to Section 7. □

**Theorem 5.** *If a graphical constant-sum game admits a fully-mixed NE $\mathbf{x}^*$ and all players employ MW updates, while the starting point $\mathbf{x}(0)$ is not a NE, then*

(a) *for any $\delta > 0$, there exists a time $T_\delta$ such that for all $t \geq T_\delta$, there exists some $i \in [m]$ and $j \in S_i$ with $x_{ij}(t) \leq \delta$. Thus, the $\omega$-set of the dynamic is a subset of the boundary;*

(b) *let $\mathcal{U}$ be an open neighbourhood of $\mathbf{x}$ which is bounded away from the boundary, the MW updates will enter $\mathcal{U}$ only finitely often, i.e., there exists a time $T$ such that for all $t \geq T$, $\mathbf{x}(t) \notin \mathcal{U}$; in other words, Poincaré recurrence property does not hold.*

Theorem 5 should be compared with Theorem 3. The key message is that although MW updates are the discrete analogue of RD, their qualitative behaviours differ significantly. The main technical reason behind is that the discretization from RD to MW updates introduces some second-order terms which accumulate in the bad way and keep *pushing* the MW updates toward the boundary. Such bad accumulation exists even when $\varepsilon$ is arbitrarily tiny, which might be surprising to people not familiar with numerical methods, since they might have the misconception that once the step-size $\varepsilon$ is sufficiently small, the discretization would always yield a good approximation of its continuous counterpart. Theorem 5(b) is a direct corollary of Theorem 5(a).

*Proof.* Suppose that $H(t)$ is bounded by some constant $q$ throughout. By Lemma 4, $H(t) \geq H(0)$ for all $t \geq 0$. Thus, for all $t \geq 0$, $\mathbf{x}(t)$ always lies in the domain

$$D := \left\{ (\mathbf{x}_1, \mathbf{x}_2, \cdots, \mathbf{x}_m) \ \middle| \ \left( -\sum_{i=1}^{m} \frac{1}{\varepsilon_i} \cdot \sum_{j \in S_i} x_{ij}^* \cdot \ln x_{ij} \right) \in [H(0), q] \right\}.$$

Consider the function $V(\mathbf{x}_1, \cdots, \mathbf{x}_m) :=$

$$\sum_{i=1}^{m} \frac{1}{\varepsilon_i} \cdot \left[ \sum_{k \in S_i} x_{ik} \cdot (\exp (\varepsilon_i \cdot u_i(\mathbf{e}_k, \mathbf{x}_{-i})) - 1)^2 - \left( \sum_{k \in S_i} x_{ik} \cdot (\exp (\varepsilon_i \cdot u_i(\mathbf{e}_k, \mathbf{x}_{-i})) - 1) \right)^2 \right].$$

We argue that $\inf_{\mathbf{x} \in D} V(\mathbf{x})$ is positive, which follows readily by the following sequence of observations. By definition of $D$, it is bounded away from the boundary. Note that $D$ is a compact set, since it is the inverse of a continuous function for some closed interval. Also, $D$ is bounded away from

any NE (this is a simple corollary of Lemma 4, since the set of NE is closed). Since for any fully mixed point $\mathbf{x}$, $V(\mathbf{x}) = 0$ if and only if $\mathbf{x}$ is a NE, and since $V$ is a continuous function on $D$, we can conclude that $\{V(\mathbf{x}) \mid \mathbf{x} \in D\}$ is bounded away from zero.

Let $\inf_{\mathbf{x} \in D} V(\mathbf{x}) = r > 0$. By Lemma 4, after $t = \left\lceil \frac{4(q - H(0))}{r} \right\rceil + 1$ steps of the MW updates, $H(t) > q$, a contradiction.

Thus, we can conclude that for any $q > 0$, there exists a time $T_q$ such that for all $t \geq T_q$, $H(t) > q$. This implies that for all $t \geq T_q$, there exists $i \in [m]$ and $j \in S_i$ such that $\frac{x_{ij}^*}{\varepsilon_i} \cdot \ln \frac{1}{x_{ij}(t)} > \frac{q}{\sum_{i=1}^m n_i}$, and hence $x_{ij}(t) < \exp\left( -\frac{\varepsilon_i q}{\sum_{i=1}^m n_i} \right)$. Theorem 5 follows by setting $q = \frac{1}{\min_i \varepsilon_i} \cdot \left( \ln \frac{1}{\delta} \right) \cdot \sum_{i=1}^m n_i$. □

## 4 Regret Lower Bound

**Theorem 6.** *In any graphical constant-sum game which admits a fully-mixed NE $\mathbf{x}^*$ and all players employ MW updates, while the starting point $\mathbf{x}(0)$ is not a NE, then there exists a sufficiently large $T$ such that for all $t \geq T$, there exists a Player $i$ (which can change w.r.t. t) with regret at least $\frac{1}{\varepsilon_i t} \cdot \ln \left( 1 + \frac{1}{2(n_i - 1)} \cdot \frac{\min_{k \in S_i} x_{ik}(0)}{\max_{k \in S_i} x_{ik}(0)} \right)$. In particular, if $\mathbf{x}_i(0)$ is uniform, then the regret is at least $\frac{1}{4\varepsilon_i (n_i - 1)t}$.*

We compare Theorem 6 with a lower bound result in Daskalakis et al. [7, Theorem 2]. Our result focuses on MW updates, and it applies to any graphical constant-sum game with a fully-mixed NE. The result in [7] focuses on a general lower bound of $\Omega(1/T)$ regret for *any* type of *distributed protocol*, which is more general than ours. To do so, they constructed a specific class of zero-sum games and show that any distributed protocol must lead to an $\Omega(1/T)$ regret in one of such games.

*Proof.* First, we show the following inequality; (*) follows from the fact $\ln(y)$ is a concave function of $y$, and a use of the Jensen's inequality.

$$
\begin{aligned}
\frac{\ln x_{ij}(T) - \ln x_{ij}(0)}{T} &= \sum_{t=0}^{T-1} \ln \frac{x_{ij}(t+1)}{x_{ij}(t)} \\
&= \frac{\varepsilon_i}{T} \sum_{t=0}^{T-1} u_i(\mathbf{e}_j, \mathbf{x}_{-i}(t)) - \frac{1}{T} \sum_{t=0}^{T-1} \ln \left( \sum_{k \in S_i} x_{ik}(t) \cdot \exp\left( \varepsilon_i \cdot u_i(\mathbf{e}_k, \mathbf{x}_{-i}(t)) \right) \right) \\
&\overset{(*)}{\leq} \frac{\varepsilon_i}{T} \sum_{t=0}^{T-1} u_i(\mathbf{e}_j, \mathbf{x}_{-i}(t)) - \frac{1}{T} \sum_{t=0}^{T-1} \sum_{k \in S_i} x_{ik}(t) \cdot \varepsilon_i \cdot u_i(\mathbf{e}_k, \mathbf{x}_{-i}(t)) \\
&= \frac{\varepsilon_i}{T} \sum_{t=0}^{T-1} u_i(\mathbf{e}_j, \mathbf{x}_{-i}(t)) - \frac{\varepsilon_i}{T} \sum_{t=0}^{T-1} u_i(\mathbf{x}(t)).
\end{aligned}
$$

Note that the final expression, when maximizing over all $j \in S_i$, is exactly $\varepsilon_i$ times the regret of Player $i$.

By Theorem 5, for $\delta = \frac{1}{2} \cdot \min_{i \in [m], j \in S_i} x_{ij}(0)$, there exists a time $T$ such that for all $t \geq T$, there exists $i \in [m], j \in S_i$ such that $x_{ij}(t) \leq x_{ij}(0)/2$. Thus, for that Player $i$, there exists some strategy $k \in S_i \setminus \{j\}$ such that $x_{ik}(t) \geq x_{ik}(0) + \frac{1}{2(n_i - 1)} \cdot x_{ij}(0)$, and hence

$$
\ln x_{ik}(T) - \ln x_{ik}(0) \geq \ln \left( 1 + \frac{1}{2(n_i - 1)} \cdot \frac{x_{ij}(0)}{x_{ik}(0)} \right) \geq \ln \left( 1 + \frac{1}{2(n_i - 1)} \cdot \frac{\min_{k \in S_i} x_{ik}(0)}{\max_{k \in S_i} x_{ik}(0)} \right).
$$

Thus, the regret of Player $i$ is at least $\frac{1}{\varepsilon_i T}$ times the RHS of the above inequality. □

Indeed, by using (1), the inequality (*) can be improved, and then we have

$$
\frac{\ln x_{ij}(T) - \ln x_{ij}(0)}{\varepsilon_i T} + \frac{1}{4\varepsilon_i T} \sum_{t=0}^{T-1} V_i(t) \leq \frac{1}{T} \sum_{t=0}^{T-1} u_i(\mathbf{e}_j, \mathbf{x}_{-i}(t)) - \frac{1}{T} \sum_{t=0}^{T-1} u_i(\mathbf{x}(t))
$$

Thus, $\frac{1}{4\varepsilon_i T} \sum_{t=0}^{T-1} V_i(t)$ can serve as a lower bound of regret. In the last section, we show that if the starting point is fully mixed but not NE, if MW updates were to stay away from the boundary, then $V_i(t)$ is bounded away from zero, and thus a regret lower bound of $\Omega_{\varepsilon_i}(1)$ could follow. But MW updates do converge to the boundary, so it is not clear how to derive a tight lower bound on this sum. In particular, we cannot rule out the possibility that MW updates converge to a subgame NE (a subgame is obtained from the original game by removing at least one strategy of some player), and if this happens, $V_i(t)$ converges to zero.

We note that essentially the same proof yields the following more general result about general games, which states that if the dynamic is not Poincaré recurrent, then the regret is at least $\Omega(1/T)$ eventually.

**Proposition 7.** *In any general game where all players employ MW updates with starting point $\mathbf{x}(0)$, if there exists an open neighbourhood $B$ around $\mathbf{x}(0)$ and a time $T$ such that for all $t \geq T$, $\mathbf{x}(t) \notin B$, then for all $t \geq T$, there exists a Player $i$ (which can change w.r.t. $t$) with regret at least $\Omega(1/T)$, where the hidden constant in $\Omega(\cdot)$ depends only on $\varepsilon$ and the radius of $B$.*

We do not have any improvement on the generic regret upper bound. In the next section, we will use such the generic bound, which is $2\varepsilon_i + \frac{C(\mathbf{x}(0))}{\varepsilon_i T}$ for Player $i$, where $C(\mathbf{x}(0))$ is a constant which depends on the starting point. (See [10].)[5] We bound this regret by $2.1 \cdot \varepsilon_i$ for all sufficiently large $T$.

## 5  Infinitely Often Almost Extinction, Infinitely Often Resurgence

In this section, we focus on two-person zero-sum (or constant-sum) games. Theorem 5 applies, i.e., beyond some finite time, there must exist some tiny probability density. A natural question to ask is will one density be tiny forever, or some densities take turn to be tiny? In this section, we prove that the former case cannot happen when $\varepsilon$ is sufficiently small. For any two-person zero-sum game $\mathcal{G}$, let $\mathsf{val}(\mathcal{G})$ denote its game value w.r.t. Player 1. In this section, we write $\varepsilon = \max\{\varepsilon_1, \varepsilon_2\}$.

Given a two-person zero-sum game $\mathcal{G}$, name the two players 1 and 2, and their strategy sets are $S_1$ and $S_2$ respectively. For $i = 1, 2$, and each $j \in S_i$, let $\mathcal{G}^{ij}$ denote the two-person zero-sum game which is formed from $\mathcal{G}$ by removing the strategy $j$ from Player $i$. Now, define

$$\theta(\mathcal{G}) := \min\left\{ \min_{j \in S_1}\left\{\mathsf{val}(\mathcal{G}) - \mathsf{val}(\mathcal{G}^{1j})\right\} \ , \ \min_{k \in S_2}\left\{\mathsf{val}(\mathcal{G}^{2k}) - \mathsf{val}(\mathcal{G})\right\} \right\}.$$

Using von Neumann's Minimax Theorem, it is easy to prove that $\theta(\mathcal{G}) \geq 0$. Intuitively this should be also clear, since removing one strategy from Player 1 will surely not benefit her, and removing one strategy from Player 2 provides less choices available to Player 2, and hence might benefit Player 1.

The following two lemmas can be easily proved using the linear program (LP) formulation of two-person zero-sum game and the Minimax Theorem; see Section 7 for their proofs.

**Lemma 8.** *If $\mathcal{G}$ is a two-person zero-sum game with a unique NE which is fully mixed, then $\theta(\mathcal{G}) > 0$.*

**Lemma 9.** *In a two-person zero-sum game $\mathcal{G}$, if both players employ MW updates, for all sufficiently large $T$, we have $\frac{1}{T} \sum_{t=0}^{T-1} u_1(\mathbf{x}(t)) \in \mathsf{val}(\mathcal{G}) \pm 2.1 \cdot \varepsilon$.*

**Theorem 10.** *In a two-person zero-sum game with a unique NE which is fully mixed, if both players employ MW updates with step-size $\varepsilon_i < \theta(\mathcal{G})/7$, then there exists at least two probability densities $x_{ij}(t)$ which exhibit the following pattern: (a) for any $\delta > 0$, $x_{ij}(t) < \delta$ for infinitely many $t$; and (b) $x_{ij}(t) \geq \theta(\mathcal{G})/7$ for infinitely many $t$.*

We give some intuition before giving the proof. By Theorem 5, there exists $T$ such that for all $t \geq T$, there must exist a density at time $t$ which is below $\delta$. Thus, it is possible to find a fixed $i$ and $j \in S_i$ such that the property (a) holds. Suppose this $i = 2$; the case $i = 1$ is symmetric. Suppose that for this $x_{ij}$, property (b) does not hold, i.e., $x_{ij}(t)$ remains below some $\kappa$ after some time $T'$. Intuitively, this implies that from time $T'$ onwards, the game essentially becomes $\mathcal{G}^{ij}$, modulo perturbation of magnitude $\mathcal{O}(\kappa)$. By Lemma 9, the long-run average payoff to Player 1 from time $T'$ onward is within the interval $\mathsf{val}(\mathcal{G}^{ij}) \pm \mathcal{O}(\kappa) \pm \mathcal{O}(\varepsilon)$. On the other hand, by Lemma 9 again, the long-run average payoff to Player 1 from time 0 onward is within the interval $\mathsf{val}(\mathcal{G}) \pm \mathcal{O}(\varepsilon)$. These two average payoffs should match, but when $\varepsilon, \kappa$ are both small, the two intervals do not overlap, a contradiction.

*Proof.* To avoid cluster of algebra, we let $v := \mathsf{val}(\mathcal{G})$ and $v^{ij} := \mathsf{val}(\mathcal{G}^{ij})$. Let $\kappa := \theta(\mathcal{G})/7$.

Suppose that the concerned density is $x_{1j}$. By the regret upper bound, for all $k \in S_1$, and for all sufficiently large $T'' > T'$,

$$\frac{1}{T'' - T' + 1} \left[ \sum_{t=T'}^{T''} u_1(\mathbf{e}_k, \mathbf{x}_2(t)) - \sum_{t=T'}^{T''} u_1(\mathbf{x}(t)) \right] \leq 2.1 \cdot \varepsilon.$$

Note that we can rewrite $\frac{1}{T'' - T' + 1} \sum_{t=T'}^{T''} u_1(\mathbf{e}_k, \mathbf{x}_2(t))$ as $u_1(\mathbf{e}_k, \frac{1}{T'' - T' + 1} \sum_{t=T'}^{T''} \mathbf{x}_2(t))$. By the Minimax Theorem, we can guarantee that there is some $k \in S_1 \setminus \{j\}$ such that $u_1(\mathbf{e}_k, \frac{1}{T'' - T' + 1} \sum_{t=T'}^{T''} \mathbf{x}_2(t)) \geq v^{1j}$. Thus,

$$\frac{1}{T'' - T' + 1} \sum_{t=T'}^{T''} u_1(\mathbf{x}(t)) \geq v^{1j} - 2.1 \cdot \varepsilon. \tag{2}$$

Next, consider Player 2. By the regret upper bound, for all $k \in S_2$, and for all sufficiently large $T'' > T'$,

$$\underbrace{u_2 \left( \mathbf{e}_k, \frac{1}{T'' - T' + 1} \sum_{t=T'}^{T''} \mathbf{x}_1(t) \right)}_{\mathcal{W}_k} + \frac{1}{T'' - T' + 1} \sum_{t=T'}^{T''} u_1(\mathbf{x}(t))$$

$$= \frac{1}{T'' - T' + 1} \left[ \sum_{t=T'}^{T''} u_2(\mathbf{e}_k, \mathbf{x}_1(t)) - \sum_{t=T'}^{T''} u_2(\mathbf{x}(t)) \right] \leq 2.1 \cdot \varepsilon.$$

Note that in the summation in the term $\mathcal{W}_k$, $x_{1j}(t) < \kappa$ by assumption. For each $t$, we construct a new probability distribution on $S_1$, denoted by $\mathbf{x}'_1(t)$, as follows: $x'_{1j}(t) = 0$, and for any $k \in S_1 \setminus \{j\}$, $x'_{1k}(t) = x_{1k}(t) + \frac{1}{|S_1| - 1} \cdot x_{1j}(t)$. Since the payoff value is always within the interval $\pm 1$, we have

$$\mathcal{W}'_k - 2\kappa := u_2 \left( \mathbf{e}_k, \frac{1}{T'' - T' + 1} \sum_{t=T'}^{T''} \mathbf{x}'_1(t) \right) - 2\kappa \leq \mathcal{W}_k.$$

By the Minimax Theorem, we can guarantee that there is some $k \in S_2$ such that $\mathcal{W}'_k \geq -v^{1j}$. Combining all the inequalities above yields

$$\frac{1}{T'' - T' + 1} \sum_{t=T'}^{T''} u_1(\mathbf{x}(t)) \leq v^{1j} + 2.1 \cdot \varepsilon + 2\kappa. \tag{3}$$

Inequalities (2) and (3) imply that the long-run average payoff to Player 1 from time $T'$ onward is within the interval $v^{1j} \pm 2.1 \cdot \varepsilon \pm 2\kappa$. But Lemma 9 states that the long-run average payoff to Player 1 from time 0 onward is within the interval $v \pm 2.1 \cdot \varepsilon$. Since the former average is obtained by only ignoring finitely many terms in the beginning, these two averages are essentially the same in the long run, i.e., the two intervals must overlap. However, this is not possible when $\varepsilon, \kappa \leq \theta(\mathcal{G})/7$.

Finally, note that among the times $t$ where $x_{ij}(t) \geq \theta(\mathcal{G})/7$, there must be another density, say $x_{i'j'}$, satisfying $x_{i'j'}(t) < \delta$ infinitely often. By reiterating the above argument for this $x_{i'j'}$, we are done. $\qquad\square$

## 6 Discussion and Some Open Problems

In this paper, we provide a better understanding of MW updates with constant step-size in graphical constant-sum games. Yet, a number of interesting problems are still unsolved. We raise some:

- While we provide a lower bound on the regret, the best upper bound we know is still the generic $\varepsilon + o_T(1)$ one, which applies in rather general scenarios and has not exploited any structure of graphical constant-sum games. Will better lower/upper bound be admissible?

- We can only prove that the fluctuating pattern described in Theorem 10 exists for two-person constant-sum games, but not general graphical constant-sum games. The technical reason is we need several nice properties of von Neumann's LP formulation and Minimax Theorem[6] to establish Lemmas 8 and 9, which are not known for graphical constant-sum games. Can we generalize?
- Even for two-person zero-sum games, Theorem 10 has not yet provided the complete picture. Will all densities exhibit such fluctuating pattern, or only some of them do? If it is the latter, given the game and the starting point, can we determine (by a mathematical proof, or by a polynomial time algorithm) which densities exhibit such fluctuating phenomenon?

## 7   Missing Proofs

**The Double Summation in the Proof of Lemma 4.**   In a graphical constant-sum game, suppose the underlying graph is $G = ([m], E)$, and each edge $(i, \ell) \in E$ corresponds to a constant-sum game; we will use the matrix $\mathbf{A}^{i\ell}$ to denote the payoffs to Player $i$ in this game.

In the calculation below, we write $x_\bullet \equiv x_\bullet(t)$, i.e., we ignore the parameter $t$.

First, we rewrite the double summation as below:

$$\sum_{i=1}^{m} \sum_{k \in S_i} (x_{ik} - x_{ik}^*) \cdot u_i(\mathbf{e}_k, \mathbf{x}_{-i}) = \sum_{i=1}^{m} \sum_{\ell:(i,\ell)\in E} (\mathbf{x}_i - \mathbf{x}_i^*)^\mathsf{T} \mathbf{A}^{i\ell} \mathbf{x}_\ell.$$

Since $\mathbf{x}^*$ is fully mixed, by the definition of NE, every entry in the vector $\left(\sum_{\ell:(i,\ell)\in E} \mathbf{A}^{i\ell} \mathbf{x}_\ell^*\right)$ must be identical. Thus,

$$\sum_{\ell:(i,\ell)\in E} (\mathbf{x}_i - \mathbf{x}_i^*)^\mathsf{T} \mathbf{A}^{i\ell} \mathbf{x}_\ell^* = 0,$$

and hence

$$\sum_{i=1}^{m} \sum_{k \in S_i} (x_{ik} - x_{ik}^*) \cdot u_i(\mathbf{e}_k, \mathbf{x}_{-i}) = \sum_{i=1}^{m} \sum_{\ell:(i,\ell)\in E} (\mathbf{x}_i - \mathbf{x}_i^*)^\mathsf{T} \mathbf{A}^{i\ell} (\mathbf{x}_\ell - \mathbf{x}_\ell^*)$$

$$= \sum_{(i,\ell)\in E} \left[ (\mathbf{x}_i - \mathbf{x}_i^*)^\mathsf{T} \mathbf{A}^{i\ell} (\mathbf{x}_\ell - \mathbf{x}_\ell^*) + (\mathbf{x}_\ell - \mathbf{x}_\ell^*)^\mathsf{T} \mathbf{A}^{\ell i} (\mathbf{x}_i - \mathbf{x}_i^*) \right]$$

$$= \sum_{(i,\ell)\in E} \Big[ \left( (\mathbf{x}_i)^\mathsf{T} \mathbf{A}^{i\ell} \mathbf{x}_\ell + (\mathbf{x}_\ell)^\mathsf{T} \mathbf{A}^{\ell i} \mathbf{x}_i \right) + \left( (\mathbf{x}_i^*)^\mathsf{T} \mathbf{A}^{i\ell} \mathbf{x}_\ell^* + (\mathbf{x}_\ell^*)^\mathsf{T} \mathbf{A}^{\ell i} \mathbf{x}_i^* \right)$$

$$- \left( (\mathbf{x}_i^*)^\mathsf{T} \mathbf{A}^{i\ell} \mathbf{x}_\ell + (\mathbf{x}_\ell)^\mathsf{T} \mathbf{A}^{\ell i} \mathbf{x}_i^* \right) - \left( (\mathbf{x}_i)^\mathsf{T} \mathbf{A}^{i\ell} \mathbf{x}_\ell^* + (\mathbf{x}_\ell^*)^\mathsf{T} \mathbf{A}^{\ell i} \mathbf{x}_i \right) \Big].$$

Note that in the final expression, there are four terms, while each term is the sum of payoffs to the Players $i$ and $\ell$ in the two-person constant-sum game corresponded by the edge $(i, \ell)$ assuming the players are using some mixed strategies. Therefore, the four terms are equal, and thus the overall expression is zero.

**Proof of Lemma 8.**   We prove the case when Player 1 has strategy $j$ removed; the case for Player 2 is symmetric.

The game value $\mathsf{val}(\mathcal{G})$ can be described to be the following: Player 1 picks a probability distribution $\mathbf{x}_1^*$ such that no matter what Player 2's choice $\mathbf{x}_2$ is, $u_1(\mathbf{x}_1^*, \mathbf{x}_2)$ is always at least $v$, and $\mathsf{val}(\mathcal{G})$ is the maximum possible value of $v$, while $\mathbf{x}_1^*$ forms the mixed strategy of the player in a NE. Due to the assumption that the unique NE is fully mixed, there is a unique fully mixed $\mathbf{x}_1^*$ that attains the maximum possible value of $v$; in other words, any $\mathbf{x}_1$ with $x_{1j} = 0$ (which is equivalent to strategy $j$ being removed) must attain a value of $v$ strictly less than $\mathsf{val}(\mathcal{G})$, i.e., $\mathsf{val}(\mathcal{G}^{1j}) < \mathsf{val}(\mathcal{G})$.

**Proof of Lemma 9.**   The proof follows closely the logic behind the derivations of inequalities (2) and (3).

**Acknowledgments**

The author would like to acknowledge Singapore NRF 2018 Fellowship NRF-NRFF2018-07 and MOE AcRF Tier 2 Grant 2016-T2-1-170. The author thanks the anonymous reviewers for their helpful suggestions and comments, and for pointing out the prior work about continuous replicator/FTRL dynamics and the interplay between them and their discrete counterparts.

## Footnotes

*Most work done while the author was at Max-Planck Institute for Informatics, Saarland Informatics Campus.

[2]In the context of game dynamics, distributive implementability means each player only needs information she can observe *locally* (e.g., payoffs to each of her own strategies) to run the updates, and does not need to know other *global* information such as the value of the underlying game matrix and the updates of other players.

[3]Now it is standard that the knowing-the-finishing-time assumption can be get rid of by employing a "doubling trick", but the step-size will be diminishing over time.

[4]Sorin [27] showed this upper bound in the unilateral setting, i.e., the bound holds for any player which uses RD, while how the environment (e.g., game payoffs, other players behaviours) varies over time does not matter.

[5]The well-known $\mathcal{O}(1/\sqrt{T})$ upper bound is indeed coming from this bound and pick $\varepsilon_i = \Theta(1/\sqrt{T})$.

[6]The root of these properties is the "absolute conflict" nature of two-person constant-sum games, which does not exist in general graphical constant-sum games.

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
