[Reviews · NeurIPS 2018]

Reviewer 1



The authors study graphical games where players play simultaneous 2-player constant-sum games with everyone they are connected to in a graph representation of player interactions. A previous result says that this class of games includes all separable zero-sum games. The authors analyze the dynamics of multiplicative weights updates: all players learn over time by multiplying weights on a strategy by a term exponential in the utility of the strategy, then normalizing over strategies. The continuous analog of multiplicative weights is replicator dynamics: weights update according to a differential equation that in some sense is the limit of multiplicative weights as the step size goes to zero. Despite the analogy, and to some surprise, the authors show that these two update rules lead to very different dynamics, even as the step size goes to zero. Replicator dynamics have properties called permanence and Poincaré recurrence that multiplicative weights do not. Multiplicative weights have a property where at least two distributions spend infinite time above a constant and also infinite time near zero. The submission seems to fit into the NIPS CFP in the category "Theory: Game Theory and Computational Economics". The authors show something interesting, surprising, and non-trivial. The paper is dense. Most of the heavy notation is necessary, and I'm glad the authors did not eliminate all proofs from the body, but some more intuition would be useful. For example, I don't really understand what "bounded away from the boundary" means in Theorem 5. What boundary? I would like to see a description in words of what Theorem 5 means. The work is worth publishing in some venue since it is natural to assume or attempt to show the multiplicative updates can get some of the nice convergence properties of replicator dynamics studied in a different literature. The authors prove a negative result to help future authors avoid this pitfall. Some questions: What is wrong with adaptive/diminishing step-size from an algorithmic perspective? If multiplicative weights fails to converge to replicator dynamics as epsilon goes to zero, is there another discrete numerical method that has a game-theoretic intuition but that does converge to replicator dynamics in the limit? Minor comments: now popular optimization perspective >> now-popular optimization perspective discrete dialogues >> discrete analogs line 140: in priori >> do you mean "a priori"? line 197: focus on... game >> focus on... games Update after author response: I still feel the paper is good and worth accepting, but I strongly request that the authors tone down claims of novelty regarding the qualitative differences between discrete and continuous dynamics, cite the sources mentioned by Reviewer #2, and make clear what is new compared to the known results references by Reviewer #2.

Reviewer 2



************* POST-REBUTTAL ******************* I gave an "accept" recommendation during the first review phase, but I was (negatively) surprised when the authors' rebuttal completely ignored part of my review, namely that Theorem 2 has already been reported in a more general, unilateral setting by Sorin (Mathematical Programming, 2009), Hofbauer, Sorin and Viossat (Mathematics of Operations Research, 2009), and subsequently extended to more general dynamics by Kwon and Mertikopoulos (Journal of Dynamics and Games, 2017) and Mertikopoulos, Papadimitriou, and Piliouras (SODA, 2018). In particular, the first and third papers above discuss extensively the interplay between continuous and discrete time - an interplay which plays an important role in the current submission as well - while the second and last papers also identify constants of motion under the replicator dynamics and discuss Poincaré recurrence. I still believe the paper should be accepted, but I've downgraded my recommendation to a "weak accept" to indicate my reservations concerning the authors' positioning relative to previous work. ************* PRE-REBUTTAL ******************* In this paper, the authors study the behavior of the multiplicative-weights algorithm with a constant step-size in zero-sum games (both standard, two-player games, and graphical games played over an interaction graph). The authors' main contributions can be summarized as follows: 1. The continuous-time limit of the multiplicative weights algorithm (known as the replicator dynamics in evolutionary game theory) incurs regret that is at most O(1/T) after time T. 2. The dynamics are "permanent" in the sense that every trajectory thereof is contained in a compact set that is disjoint from the boundary of the game's mixed strategy space (i.e. in the long term, the share of every strategy is bounded away from zero). 3. The above conclusions do not hold if the algorithm is run with a constant, positive step-size: in this case, the algorithm's regret is bounded from below as \Omega(1/(epsilon T)) where epsilon is the algorithm's (constant) step-size, and the dynamics are not permanent (nor Poincaré recurrent). The paper is well-written, clearly structured and I enjoyed reading it. However, some of the authors' contributions are not new (see below), so the overall novelty is limited. Still, the results that are new are (in my opinion) sufficient and will interest the NIPS community, hence my positive evaluation. My main criticisms are as follows: 1. The fact that the discrete analogue of the replicator dynamics has a different behavior than the continuous dynamics is not as surprising as the authors claim, see for instance the 2009 paper of Sorin in Mathematical Programming and the monograph of Benaïm (1999) on stochastic approximations, where it is made clear that constant step-size discretizations of dynamical systems could have a wildly different behavior than their continuous-time counterparts. 2. The fact that the replicator dynamics (i.e. the continuous-time limit of the multiplicative weights algorithm) lead to O(1/T) regret was first reported by Sorin (Mathematical Programming, 2009) and Hofbauer, Sorin and Viossat (Mathematics of Operations Research, 2009), and subsequently extended to all mirror descent / follow-the-regularized-leader dynamics by Kwon and Mertikopoulos (Journal of Dynamics and Games, 2017) and Mertikopoulos, Papadimitriou and Piliouras (SODA, 2018). In fact, this O(1/T) bound does not even require a game: it holds for arbitrary (measurable) series of payoffs, even if those are not coming from a game. 2. The permanence result for the replicator dynamics is also alluded to in the classical book of Hofbauer and Sigmund and the more recent paper of Hofbauer, Sorin and Viossat (2009). [Granted, these works do not mention graphical zero-sum games, but the essence of the result is there and the proof strategy is exactly the same] See also the SODA 2018 paper of Mertikopoulos, Piliouras, and Papadimitriou where the authors prove a more general 4. In general, why would one want to employ the multiplicative weights algorithm with a constant step-size? As the authors state themselves, achieving no-regret in an anytime setting requires a decreasing step-size anyway (either due to a doubling trick or a sqrt{t} schedule). It is also well-known that a decreasing step-size is much better suited to stochastic environments (which are the norm in any randomized action selection framework), so it is not clear why one would want to use a constant step-size algorithm in the face of uncertainty (either temporal or informational). There are some further minor issues (such as the requirement that epsilon ≤ 1/4), but these are not important enough to raise here.

Reviewer 3



Summary: The paper presents several properties of the convergence process of replicator dynamics (RD) and multiplicative weights (MW) updates in the class ot graphical constant-sum games, which generalizes two-person zero-sum games. It shows that even though one is a discrete analogue of the other, the convergence properties differ substantially. While the action probabilities in RD stay in constant distance form zero, in MW, they infinitely often are arbitrarily close to zero. Quality: The claims in the paper seem to be sound. The paper builds on the previous work and provides sound theoretical arguments. The paper would be stronger if it discussed possible impact of the presented theoretical findings. They certainly help us get deeper understanding of how th convergence looks like, but it is not obvious that this understanding will have some practical impact. Originality: I have only limited knowledge of the field, so it is hard for me to assess originality of the results. However, I was not familiar with the presented results before and the presented proofs are not trivial. Significance: The paper could have made a better job in explaining the significance of the results. I consider the oscillating behavior of MW on the boundary after a finite number of iteration to be the most interesting result. However, this relays on the constant step size, which is not the most common setup in practical applications. The other results may and may not be useful for future research and better understanding of the algorithms. However, this is often the case with purely theoretical research. Clarity: The paper is well written and quite easy to follow. I provide some minor suggestions to improve readability below. Detailed comments: 13-15: In the last sentence of the abstract, it was initially not clear to be what exactly is meant by probability densities getting "above h infinitely often". Consider rephrasing the sentence. 22: I am not sure what "distributive implementability" means. A sentence of explanation would be helpful. 40: "in priori" -> "a priori" 44: requires -> require 107: In the differential equation for RD, "u" should be "u_i" 134: The modified H should have a different name for clarity, such as "H'" 161: remove "Consider the domain"